Subject Category:
Biology (whole organism)

Subject Areas:
behaviour/physiology/developmental biology

Keywords:
coping styles, personality, life history, hypoxia, stress, cortisol

Author for correspondence:
Børge Damsgård
e-mail: borge.damsgard@unis.no

# Proactive avoidance behaviour and pace-of-life syndrome in Atlantic salmon

Børge Damsgård[1,2], Tor H. Evensen[2], Øyvind Øverli[3], Marnix Gorissen[4], Lars O. E. Ebbesson[5], Sonia Rey[6] and Erik Höglund[7,8]

[1]The University Centre in Svalbard (UNIS), 9171 Longyearbyen, Norway
[2]Nofima, 9291 Tromsø, Norway
[3]Department of Food Safety and Infection Biology, Norwegian University of Life Sciences, 0454 Oslo, Norway
[4]Department of Animal Ecology and Physiology, Institute of Water and Wetland Research, Radboud University, 6525AJ Nijmegen, The Netherlands
[5]Uni Research, 5020 Bergen, Norway
[6]Institute of Aquaculture, University of Stirling, FK9 4LA Stirling, UK
[7]Center of Coastal Research, University of Agder, 4604 Kristiansand, Norway
[8]Norwegian Institute of Water Research, 0349 Oslo, Norway

BD, 0000-0002-7731-0168; SR, 0000-0002-3406-3291; EH, 0000-0002-1350-8255

Individuals in a fish population differ in key life-history traits such as growth rate and body size. This raises the question of whether such traits cluster along a fast-slow growth continuum according to a pace-of-life syndrome (POLS). Fish species like salmonids may develop a bimodal size distribution, providing an opportunity to study the relationships between individual growth and behavioural responsiveness. Here we test whether proactive characteristics (bold behaviour coupled with low post-stress cortisol production) are related to fast growth and developmental rate in Atlantic salmon, *Salmo salar*. Boldness was tested in a highly controlled two-tank hypoxia test were oxygen levels were gradually decreased in one of the tanks. All fish became inactive close to the bottom at 70% oxygen saturation. At 40% oxygen saturation level a fraction of the fish actively sought to avoid hypoxia. A proactive stress coping style was verified by lower cortisol response to a standardized stressor. Two distinct clusters of bimodal growth trajectories were identified, with fast growth and early smoltification in 80% of the total population. There was a higher frequency of proactive than reactive individuals in this fast-developing fraction of fish. The smolts were associated with higher post-stress plasma cortisol than

parr, and the proactive smolts leaving hypoxia had significant lower post-stress cortisol than the stayers. The study demonstrated a link between a proactive coping and fast growth and developmental ratio and suggests that selection for domestic production traits promotes this trait cluster.

# 1. Introduction

Associations between individual variation in life history and behaviour are increasingly documented, e.g. [1–3]. According to this, the pace-of-life syndrome (POLS) hypothesis predicts that behavioural traits, showing consistency between context and over time, are related to life-history traits. More specifically, bold and more aggressive individuals are predicted to develop faster and die younger than shyer and less aggressive individuals [4]. This type of behavioural trait correlation is well established within the concept of proactive and reactive stress coping styles. In addition, the stress response in proactive individuals is dominated by active avoidance and high sympathetic reactivity. In mammals, this is linked to a low hypothalamus–pituitary–adrenal (HPA), and in teleosts a low interrenal (HPI) axis responsiveness. The reactive individuals are on the other hand characterized by decreased mobility, high HPA/HPI axis responsiveness and low sympathetic reactivity.

In general, salmonids show a high degree of life-history trait variability. For example, in Atlantic salmon, *Salmo salar*, the timing of when larvae emerge from a spawning nest to establish a feeding territory can vary by up to several weeks. In accordance with POLS, studies in the early 1990s showed that larvae emerging early from the spawning nests were more aggressive and bolder than larvae emerging later [5,6]. Moreover, later in the life history, juveniles leave the nursery creek and migrate out to the sea after a smoltification process, including behavioural, physiological and morphological changes prior to the migration. The timing of this ontogenetic shift normally varies between 1 and 5 years post-hatching, and each preadaptation is regarded as a state-dependent individual life-history decision and includes changes in growth pattern [7–9].

That developmental ratio is related to stress coping styles is further confirmed by a study on larvae from farmed rainbow trout, *Oncorhynchus mykiss*, strains selected for high (HR) or low (LR) post stress plasma cortisol response [10], resembling the reactive and proactive stress coping styles respectively [10–13]. In the latter study, larvae originating from the LR proactive strain reached time to emerge earlier than the reactive strain. However, studies on the relationship between stress coping styles and larval emergence time show elusive results in natural and farmed populations of salmonid fishes [14–17]. Still, whether time to reach later ontogenetic shifts is related to stress coping styles in salmonid fish remains to be investigated.

In teleosts, a number of studies in different species suggest that contrasting behavioural responses to hypoxia are related to stress coping styles [18–20]. Following this, [21] used the HR/LR rainbow trout strains to develop a method for sorting fish by stress coping styles using hypoxia avoidance. Since then, this method has been used to sort fish according to stress coping styles in other fish species [16,22], suggesting that a relationship between coping styles and responses to hypoxia is a widespread phenomenon within this animal group.

Identification of proximate and ultimate causes of variation in fast-slow POLS has become a pivotal topic in ecology and evolutionary biology, mapping how key life-history parameters determine fitness in wild or farmed populations. In this study, stress coping style was characterized by hypoxia avoidance and HPI reactivity (quantified as post-stress cortisol) in fractions of Atlantic salmon showing bimodality in growth trajectories and time to reach smoltification, to further investigate the relation between stress coping styles and developmental rate in salmonid fish. Our main hypothesis is that stress coping styles are related to developmental rate, linking the behavioural and physiological mechanisms in the pace-of-life syndrome.

# 2. Material and methods

## 2.1. Rearing and tagging of the fish

The study was conducted at the NOFIMA Aquaculture Research Station in Tromsø, northern Norway, using 0+ Atlantic salmon of the AQUAGEN breed (Atlantic QTL-innova IPN). The fish

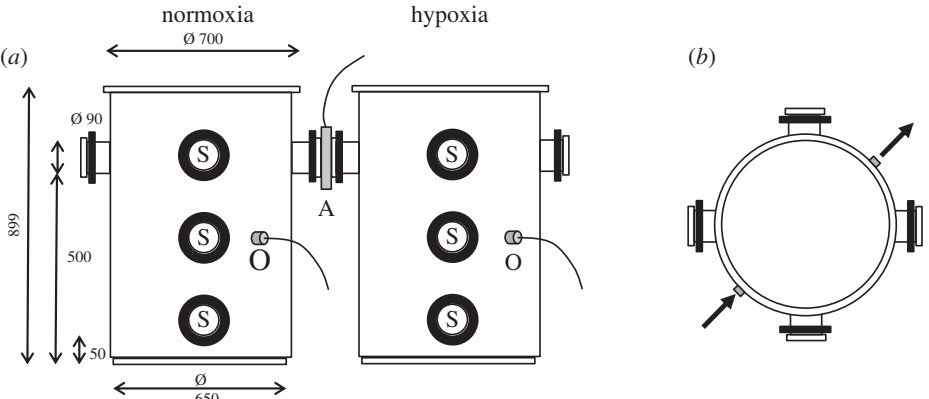

**Figure 1.** Experimental set-up for normoxia and hypoxia tanks. (*a*) Side view and (*b*) Top view. The symbol 'A' indicates the location of the PitTag antenna, 'O' the oxygen sensors, 'S' the infrared activity sensors, and the arrows indicate the water inlets and outlets.

eggs were transferred to the research station on 25 January 2012, hatched on 6 February and feeding commenced on 19 March (mean body mass of the fish at that time was 0.18 g). After first artificial feeding, fish were reared at 10°C water temperature, continuous light and surplus feeding (Skretting Nutra). The fish population ($n = 480$) was randomly divided into eight groups of 60 fish each, and reared in circular tanks (diameter 68 cm, depth 32 cm; volume 116.2 l) with flow-through fresh water. The groups were treated equally throughout the experiment. To keep the size variability within each tank, the fish were not sorted at any time. The fish were individually tagged using internal 12 mm PitTags (HPT12 tags in pre-loaded tray, Biomark, Boise, US), injected with a MK-25 implant gun. All fish were weighed at intervals throughout the growth period (age approx. 7, 9, 11 months post hatching), and at the last measurement prior to the stress experiment the mean fish body mass was 57.1 g, ranging from 7.4 to 118.8 g ($n = 471$). The total mortality during the experiment was 9 fish (less than 2%). The fish population developed a bimodal size-frequency distribution before seven months of age: a lower modal group with slow-growing fish, and a higher modal group with fast-growing fish, denoted as lower modal (LM) and upper modal (UM), respectively (figure 2*a,b*).

## 2.2. Behavioural hypoxia avoidance experiment

The set-up for behavioural experiments consisted of two custom-made circular tanks (inner diameter 65 cm, water depth 60 cm; 199 l; Cipax AS, Bjørkelangen, Norway, figure 1*a,b*). The tanks were connected at the water surface level with a tube (inner diameter 9 cm), integrated with a custom-made spool PitTag antenna (Biomark Ltd, Boise, US), and linked to a Biomark FS2001 reader and tag manager software. Each tank had separate water in- and outlets, with a water flow of 3.5 l min$^{-1}$. The inlet in the hypoxia tank was connected to a gas exchanger (diameter 29 cm, depth 125 cm; 82 l). In the exchanger, N$_2$ gas (15 mg l$^{-1}$) deoxygenated the incoming water, without any other changes in water flow or exchange. The average oxygen levels (mg O$_2$ l$^{-1}$) in the tanks were monitored continuously using an O$_2$-sensor (Oxyguard MINI-DO galvanic O$_2$ probe) mounted to the tank wall at 30 cm depth, and the data were stored using the AutoResp 1 program (Loligo Systems, Tjele, Denmark). Control tests prior to the experiment demonstrated that the oxygen depletion was homogeneous throughout the water body in the hypoxia tank. The swimming activity of the fish was monitored using three infrared sensors (RL28-8-H-2000-IR, Pepperl+Fuchs GmbH, Mannheim, Germany) in each tank, mounted outside of glass windows, close to the surface, at 30 cm water depth, and close to the tank bottom, respectively. The sensor recorded fish passing a narrow (2°), invisible light beam (880 nm) with a 2 ms response time, and a custom-made computer program (Spider AS, Tromsø, Norway) was used to calculate the number of times a fish passed the light beams per minute (which served as an estimation of the relative activity), as well as the relative swimming depth of the fish population. This was used to estimate the mean swimming depth of the whole population (cm above the bottom) as function of the O$_2$ saturation. Two video cameras were used to observe the fish passing the tube between the tanks.

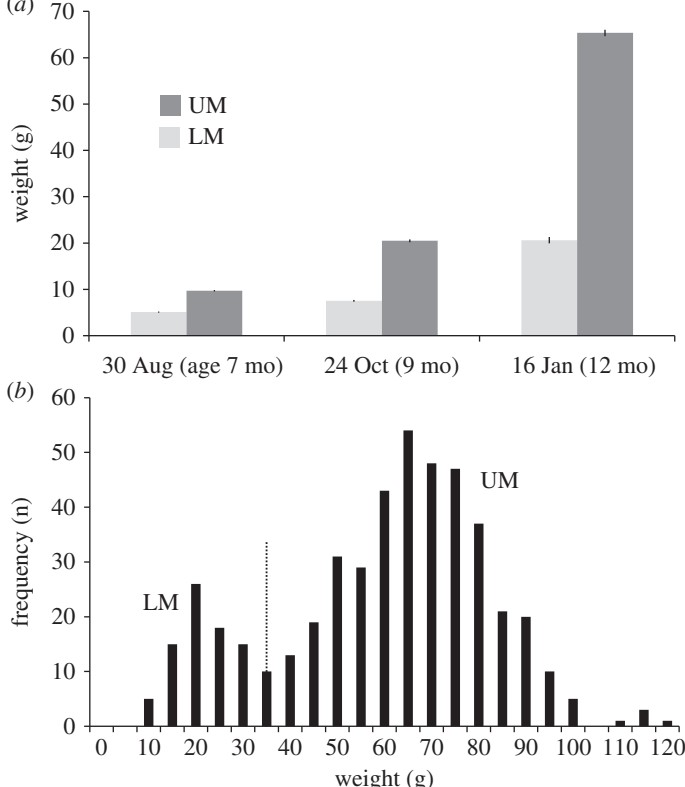

**Figure 2.** Development of bimodal growth pattern in Atlantic salmon, *Salmo salar* ($n = 471 – 478$). (a) Mean body mass $\pm$ s.e.m. (range $0.12 – 0.69$), in slow-growing lower modal (LM) and fast-growing upper modal (UM) from hatching to endpoint sampling, (b) Weight frequency distribution at endpoint sampling. The dotted line represents the separation between LM ($n = 89$) and UM ($n = 382$).

## 2.3. Hypoxia test

The hypoxia tests were conducted between 30 October and 9 November 2012 (approx. nine months post hatching), and the eight tanks were tested in random order. Each test took approximately 5 h and started at 08.30. All tests were conducted equally. Prior to each test, the tanks were cleaned, water temperature regulated if necessary, and the water flow in each tank set to $3.5 \, l \, min^{-1}$. The air pressure was measured (mm Hg, YSI Professional Pro-DO, Yellow Spring, USA) and used to continuously estimate oxygen concentration ($mg \, O_2 \, l^{-1}$), and partial pressure of oxygen ($pO_2$). Prior to the decline of the oxygen levels, oxygen concentration ranged from $10.7 – 10.9 \, mg \, O_2 \, l^{-1}$ ($98.3 – 100.4$ kPa), with a water temperature of $10.9 – 11.3°C$. The decline in oxygen concentration was similar in all eight tests, and after the onset of hypoxia, the hypoxic tank showed a nearly linear reduction in oxygen levels of about $0.05 – 0.10 \, mg \, O_2 \, min^{-1}$ (figure 3a), while the normoxic tank was not affected. The endpoint for each experiment was set to $2.5 \, mg \, O_2 \, l^{-1}$, corresponding to 25% saturation of oxygen, using AutoResp's set point regulation of the $N_2$ gas valve (Loligo LDAQ, Loligo, Denmark).

The fish were caught with a dip net and transferred to the experimental set-up as carefully as possible and were released in the hypoxia tank. The two tanks were left undisturbed behind an opaque curtain during the test. The monitoring of the fish started at once, and the fish were allowed to acclimate in the system for 3 h prior to the change in the oxygen level. During the decline in oxygen, the water flow in the hypoxia tank was directed through the gas exchanger, and a door between the hypoxia tank and the normoxia tank was opened so fish could swim freely between the tanks. When the oxygen level in the hypoxia tank reached 25% $O_2$ saturation, the experiment was terminated, and the fish transferred back to the holding tanks. The oxygen decline took typically 1.5 h after onset of the gas exchanger (average 86.6 min down to 24.5% oxygen, $n = 8$). The fish leaving hypoxia were classified as Leavers, while the ones staying in the hypoxia tank during the decline in oxygen concentration were classified as Stayers.

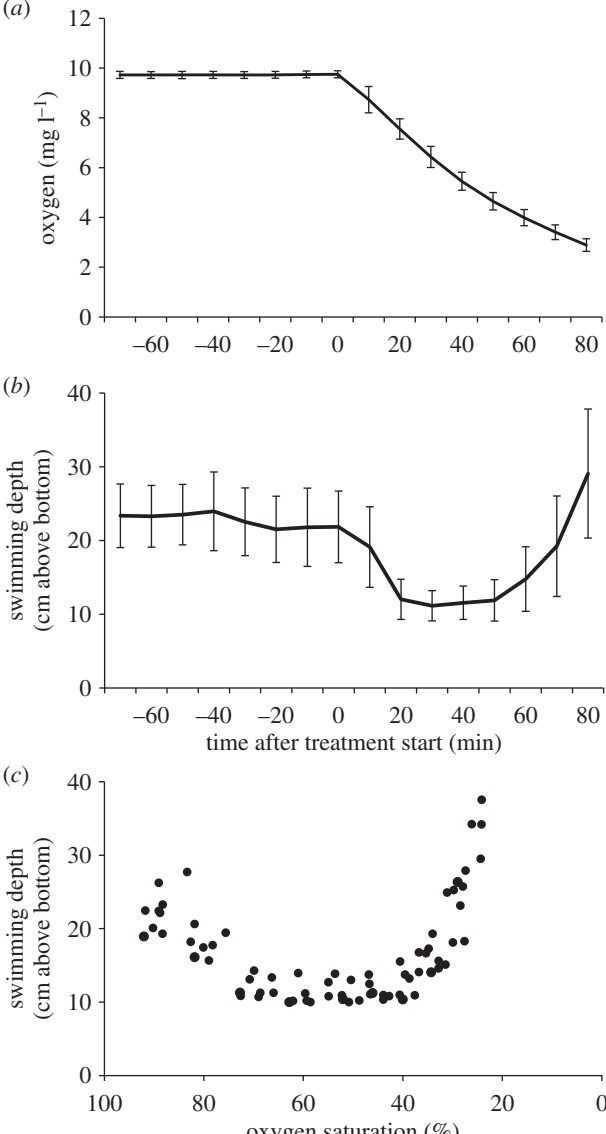

**Figure 3.** Hypoxia tests with Atlantic salmon, *Salmo salar* ($n = 471$). (*a*) Mean oxygen concentration (mg $O_2$ $l^{-1}$) in the hypoxia tank in 10 min time units, 80 min before and after the onset of hypoxia (solid line, $\pm$ s.e.m.), (*b*) Mean swimming depth (cm over bottom of the tank, $\pm$ s.e.m.) in the same time unit as (*a*), and (*c*) Mean swimming depth of the whole fish population (cm over bottom of the tank) as a function of oxygen saturation (% $O_2$).

## 2.4. Acute stress test and cortisol analyses

After the hypoxia experiment, fish were reared as prior to the experiment, and more than two months later, on 22 January (approx. 11 months post hatching), a confinement stress test was conducted sequentially to five randomly chosen fish groups. During the stress test, a subsample of 16 fish was quickly netted out of each of the five tanks (unstressed control group, $n = 80$). The water level in each tank was then lowered to 5 cm without changing the water flow, and the fish were left in this confinement for 30 min. After that, the water level increased to normal, and 30 min later 16 fish were sampled from each tank (stressed group, $n = 80$). The onset of the sampling was sequential, so each fish group was sampled at different times. After sampling, controls and stressed fish were treated equally. Fish were euthanized with an overdose (0.1% v/v) buffered MS-222 (Finquel®, Argent Chemical Laboratories, Redmond, USA), the body mass and fork length recorded, and blood was sampled from the caudal vessels using 23G needles fitted on 1-ml heparinized syringes. The samples were centrifuged for 5 min (3800 rpm at 4°C), and blood plasma stored at −80°C until further analyses. Plasma cortisol was measured in duplicates using a radioimmunoassay in a 96-well plate according to [23]. The antibody used shows 100% cross reactivity with cortisol, 0.9% with

11-deoxycortisol, 0.6% with corticosterone, and less than 0.01% with 11-deoxycorticosterone, progesterone, 17-hydroxyprogesterone, testosterone and oestradiol. Inter- and intra-assay variations were 12.5 and 3.5%, respectively.

## 2.5. Seawater challenge test

To test the smoltification status of the fish, a standard seawater challenge test [24] was conducted on 21 March (approx. 13 months post hatching) and included 10 freshwater controls (prior to the seawater challenge treatment) and 62 fish post-treatment. Without changing any other rearing conditions, the fresh water in the tank was replaced with seawater for 24 h, and approximately 1 ml blood was sampled from the caudal vessels using heparinized syringes, centrifuged (3800 rpm, 4°C) for 5 min, and stored at −80°C until further analyses. Plasma chloride concentration was analysed in duplicate, using a Coring Chloride Analyzer 925 (Medfield, USA).

## 2.6. Statistics

If not specified otherwise in the text, values are presented as mean ± s.e.m. Two distinct responses to hypoxia were observed; fish that stayed (Stayers) and left (Leavers) hypoxic conditions. The frequency of Stayers and Leavers in the slow and fast growers was analysed using a chi-square test. Cortisol data were analysed using two-way analysis of variance (ANOVA) followed by Sidak's *post hoc* multiple comparison tests (with adjusted α-level) in case of a significant interaction (behaviour responses to hypoxia × stress) effect. Because of the low number of Lower Modal (LM) fish ($n = 2$) in the control group, we were not able to perform further statistical analysis investigating potential differences between the LM and Upper modal (UM) groups in plasma cortisol. All statistical analyses were carried out using GraphPad Prism (version 6 for Mac, GraphPad Prism Software, La Jolla, CA, USA).

# 3. Results

## 3.1. Growth and development

During the growth period prior to the experiment, the eight groups of fish developed a bimodal growth pattern, with 18.9% of the fish in the lower modal group (LM), and 81.1% in the fast-growing group (UM), the body mass at the junction of the two groups being approximately 35 g (figure 2b). The growth separation was evident already at the age of seven months post hatching, and at the time of the stress test the bimodality consisted of slow-growing fish (LM) on $20.5 \pm 0.7$ g ($n = 89$), and UM fish of $65.7 \pm 0.8$ g ($n = 382$). The seawater challenge test confirmed that UM fish were physiological smolts with an average plasma chloride of $139.6 \pm 1.0$ mmol l$^{-1}$ ($n = 47$), comparable to the freshwater controls of $126.9 \pm 1.3$ mmol l$^{-1}$ ($n = 10$), while the LM fish where not smoltified with a plasma chloride level of $178.5 \pm 8.1$ mmol l$^{-1}$ ($n = 15$).

## 3.2. Behavioural responses to hypoxia

Prior to the onset of hypoxia the mean swimming depth varied between 20 and 30 cm above the bottom of the tank (figure 3b). Within 20 min after the onset of the oxygen decline the general behavioural response was that the fish became inactive and located close to the bottom of the tank. This behavioural response lasted until 50–60 min after the onset, when some of the fish became hyperactive and swam close to the water surface (figure 3b). These behaviours were confirmed using the video observations and were consistent in all eight tests. When the mean swimming depth of the whole population was calculated as a function of oxygen saturation, it became clear that the fish became increasingly inactive during the decline from 100 down to approximately 70% oxygen saturation. Fish became hyperactive at oxygen saturation less than 40% (figure 3c) and some of the fish positioned close to the water surface, and actively sought a way out of the tank to the connected normoxia tank. In total, 264 out of 471 fish (56%) were defined as Leavers, ranging from 19 to 42 fish per tank. The Leavers swam into the normoxic tank on average after 69.4 min, with an oxygen level of 3.3 mg O$_2$ min$^{-1}$ (30.4% saturation). Most of the LM fish (80%) did not leave the hypoxic tank during the oxygen decline, while 53% of the UM fish were Leavers (figure 4a). There was a significant

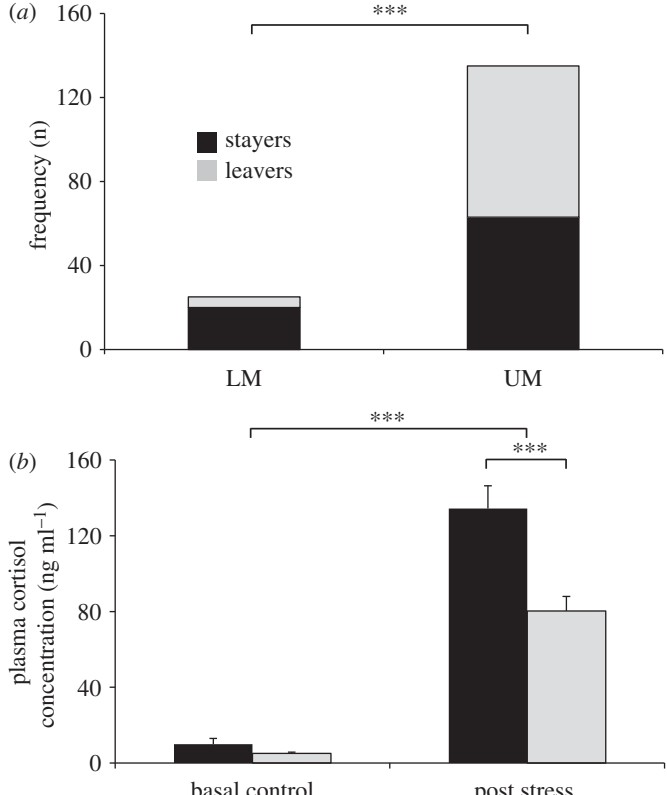

**Figure 4.** (a) Number of Stayers (Black bars, $n = 207$) and Leavers (Grey bars, $n = 264$) in the hypoxia tests in non-smoltified LM and smoltified UM Atlantic Salmon, *Salmo salar*. *** denotes significant difference ($p < 0.001$) in the frequencies of Leavers and Stayers. (b) Plasma cortisol concentration (ng ml$^{-1}$) before (basal control) and after an acute confinement stress in UM Stayers (Black bars, $n = 63$) and UM Leavers (Grey bars, $n = 69$) in the hypoxic tests. *** denotes significant difference ($p < 0.001$) in the plasma concentration between control and stress treatment, and between Leavers and Stayers.

higher frequency of Leavers in UM than in LM (chi-square 27, $p < 0.001$), and no size differences between Leavers and Stayers UM fish.

## 3.3. HPI axis reactivity of Leavers and Stayers

The mean cortisol concentration increased significantly ($U = 59$; $p < 0.0001$) after the confinement stress from $7.0 \pm 1.2$ ng ml$^{-1}$ ($n = 80$) prior to the stress period, to $100.6 \pm 7.4$ ng ml$^{-1}$ ($n = 79$) 1-h post-stress. The LM fish ($60.3 \pm 18.7$ ng ml$^{-1}$, $n = 12$) had a significant lower ($U = 163$; $p = 0.0007$) cortisol level post-stress compared to the UM fish ($107.8 \pm 7.8$ ng ml$^{-1}$, $n = 67$, figure 4b). A significant interaction effect was observed in UM fish (hypoxia test × stress test: $F_{1,131} = 11.49$; $p = 0.0009$). The UM Leavers had a significantly lower cortisol level ($80.3 \pm 7.6$ ng ml$^{-1}$, $n = 34$; $p < 0.0001$), compared with the UM Stayers ($134.4 \pm 11.9$ ng ml$^{-1}$, $n = 33$). The LM fish showed a less distinct difference between Leavers and Stayers, and the difference could not be tested due to a low number of Leavers. There were no significant differences between Leavers and Stayers in baseline cortisol ($p < 0.99$).

## 4. Discussion

Two distinct growth trajectories were observed in our population of Atlantic salmon. Moreover, the seawater challenge test confirmed an earlier smoltification in fast growers (UM; upper modal) compared to the slow growers (LM; lower modal). This is consistent with the bimodality of life-history traits demonstrated previously in a number of laboratory and field studies in most salmonids [7,25,26]. The smoltification process includes a suite of behavioural, neural, physiological and morphological changes, preparing the fish for a life in seawater. The environmental light conditions are believed to be the most important cue to start and synchronize this process, e.g. [27–30]. In nature, the smoltification process takes place in the second to fifth year of life, whereas

contemporary farmed salmon are induced to smoltify (via manipulation of light conditions) in their first year of life [29].

A higher proportion of individuals left hypoxic conditions in fast growers than in slow growers in the present study. One proximate explanation for this may be a higher metabolic rate and body size in smolting fish, and thus a lower threshold for behavioural responses to declining oxygen [31,32]. A review of the relationship between fish species size and hypoxia threshold concludes that fish size *per se* had little impact on hypoxia tolerance [33]. In addition, there were no size differences between Leavers and Stayers, and few of the slow-growing fish left hypoxia even at 25% oxygen saturation. In order not to harm our test fish we terminated the experiments at 25% saturation, as our previous experience indicated that Stayers would not leave hypoxia, even if oxygen saturation were lowered to lethal levels. Behaviourally this is comparable with events of e.g. low food availability or suboptimal environmental conditions; the individuals have the option of moving to other habitats or wait out the situation. In a study of Atlantic cod, *Gadus morhua*, behaviour in hypoxia, the swimming speed increased at moderate oxygen decline, and then decreased by greater than 40% at progressive deep hypoxia, and [34,35] propose that the decline is an adaptation to survive habitats with low oxygen. Previous studies in sea bass, *Dicentrarchus labrax*, Gilthead sea bream, *Sparus aurata*, and Atlantic salmon demonstrates that a faster escape response from hypoxia is associated with risk taking behaviour and boldness [16,22,36]. Taking our results and the aforementioned studies into consideration, this suggests a positive relationship between boldness and developmental rate in the present study. This is in line with the POLS hypothesis, predicting that proactive/bold personality types show higher mortality than shy/reactive personality types, and compensate this by a higher growth rate, developmental rate and fecundity [4,37]. By this, both extremes in the proactive/bold-reactive/shy continuum are expected to reach the same lifetime fitness. However, while there is rather extensive literature on the relationships between personality type growth and/or fecundity, the information on how these factors are related to developmental rate is meagre, for references see [38]. Yet, there are a few studies in salmonid fishes supporting that personality types are related to developmental rate, as for example suggested by [6] in Atlantic salmon larvae emerging from the spawning ground. This relationship is further strengthened in a study using the selected HR and LR rainbow trout lines, showing proactive and reactive behavioural characteristics respectively, where larvae from the proactive HR strain emerged from the spawning nest earlier than the reactive LR strain [10]. In the present study, individual variation in growth and developmental rate were associated with behavioural responses to hypoxia, suggesting a link between stress coping styles and the time to reach the ontogenetic niche shift to smoltify and migrate to the sea. In aquaculture, selecting for highly competitive fast-growing proactive individuals is common [39]. The occurrence of such contrasting behavioural traits in domesticated lines raises the question for future studies of whether successful expressions of fast-proactive phenotypes depend on the presence of at least a small fraction of passive copers in the population.

Differences in energy demands have been proposed as a proximate explanation behind behavioural differences between individuals [4,40,41]. Particularly, the energetic maintenance cost of an individual's resting metabolic rate has been strongly associated with individual differences in behaviour. Accordingly, POLS predict that bold, fast-growing and developing individuals should be associated with higher metabolic costs [38,42,43]. In line with this, different energetic strategies have been suggested to underlie individual variability in the behavioural response to hypoxia. A high locomotor activity response during hypoxia is associated with active hypoxia avoidance, individuals showing low locomotor activity are characterized by lower oxygen consumption, i.e. a sit and wait coping strategy [18]. Thus potentially, metabolic differences and the smoltification process may underlie the contrasting responses to hypoxia in the present study. However, studies in the HR-LR rainbow trout lines suggest limited impact of metabolisms on differences in behavioural responses to hypoxia [44,45]. Further studies are needed to reveal if individual variation in metabolic rates shows constancy throughout ontogeny and if this is related to coping styles/personality, as predicted by POLS.

Generally, the proactive behavioural profile has been characterized by a high level of active avoidance together with a low HPI axis reactivity, usually understood as an active attempt to counteract the stressful environment [46]. Reactive coping style, on the other hand, usually involves immobility and high HPI axis activity [47]. This is very well in accordance with our result, suggesting that the Leavers left at an oxygen level of approximately 40% proactively try to avoid the stressful situation. Moreover, these individuals also showed lower post-stress plasma cortisol levels compared to stayers, further confirming that Stayers and Leavers correspond to the proactive and reactive stress coping style, respectively. Similarly, [18] reported active hypoxia avoidance in rainbow trout with low HPI axis

reactivity. However, studies in the HR-LR lines demonstrate contrasting results, showing that the reactive HR line was associated with active avoidance to hypoxic conditions [21]. It has become increasingly clear that the link between behavioural and stress reactivity axis of the pro- and reactive coping styles is less rigid that previously thought. For example, [48] suggested that individuals may vary within the coping style behavioural reaction and stress axis independently. In line with this [10] suggested some of the contrasting characteristics between the HR-LR lines may be independent of the neuroendocrine stress axis. The present study shows that active avoidance of hypoxia is associated with low HPI activity. This, together with a study performed by [16] demonstrating that hypoxia avoidance is associated with social dominance and aggression, lends support to a relationship between hypoxia avoidance stress coping style and HPI axis activity in Atlantic salmon. The present results are in accordance with studies in sea bass and sea bream, suggesting that sorting fish with respect to hypoxia avoidance may offer a high throughput method of sorting fish in respect to stress coping styles. Moreover, it is important to point out that smoltification is associated with neural and endocrine changes rather early in the development [27,49], and it cannot be excluded that the differences in frequencies of Stayers and Leavers within slow and fast growers may be related to such differences.

# 5. Conclusion

In conclusion, fast growing Atlantic salmon reached the ontogenetic niche shift, smoltification, before slow-growing individuals. Moreover, these individuals also showed proactive characteristics, including active hypoxia avoidance and low HPI-axis post-stress reactivity. Taken together, this supports the POLS concept, including a positive relationship between boldness/proactivity, growth and developmental rate. In conclusion, it appears that the rapid selection for domestic production traits that occurs in salmon conserve or promote pace-of-life syndromes and associated physiological/ behavioural trait variation.

Ethics. The work was performed under the approved protocol entitled 'Behavioural sorting in Atlantic salmon' at the Aquaculture Research Station in Tromsø (H12/29/006.1/04.06.12/KuF), Norway, providing authorization according to the Norwegian 'Forskrift om bruk av dyr i forsøk' (regulation of the use of animals in experiments), under the Animal Welfare Act. Project leader B.D. has approved course in animal experimentation (class C) according to the Norwegian Research Animal authority, and all personnel handling animals had approved experiment licences.

Data accessibility. The data from the experiment have been uploaded to Dryad Digital Repository at: http://dx.doi.org/10.5061/dryad.4187519 [50].

Authors' contributions. B.D. was mainly responsible for the design of the study, the experimental work and acquisition of the data and drafting of the manuscript, T.H.E. participated in the design and the data acquisition, Ø.Ø. participated in the acquisition of the data, M.G. participated in the acquisition of the data and cortisol analysis and statistical analysis, L.O.E.E. participated in the acquisition of the data, S.R. participated in the acquisition of the data and drafting of the manuscript, and E.H. participated in the acquisition of the data and drafting of the manuscript. All authors contributed to the design, acquisition and drafting of the manuscript and all authors gave final approval for publication.

Competing interests. We have no competing interests.

Funding. This study was a part of the task SALMONCOPE within the project 'COPEWELL', supported by the European Union under the 7th Framework Programme FP7-KBBE-2010-4 Contract no.: 265957, which funded all participants, and experimental costs of all authors.

Acknowledgements. We would like to acknowledge the staff of the Aquaculture Research Station in Tromsø, Norway, for technical assistance during the experiment, to Tom Spanings (Radboud University, The Netherlands), to Guro Sandvik and Christine Couturier (University of Oslo, Norway), and to Per-Ove Thörnqvist (Uppsala University, Sweden) for help with experimental design and sampling, and to Thamar Pelgrim (Radboud University) for analysis of plasma samples.

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
