## [Reviewer comments · Royal Society Open Science]

Review History

RSOS-181859.R0 (Original submission)

Review form: Reviewer 1

Is the manuscript scientifically sound in its present form?

Yes

Are the interpretations and conclusions justified by the results?

Yes

Is the language acceptable?

Yes

Is it clear how to access all supporting data?

Yes

Do you have any ethical concerns with this paper?

No

Have you any concerns about statistical analyses in this paper?

No

Recommendation?

Major revision is needed (please make suggestions in comments)

Comments to the Author(s)

I have reviewed the manuscript "Proactive avoidance behaviour and pace of life syndrome in Atlantic salmon". The goal of this study was to investigate whether there is a relationship between growth and development rate and boldness in behaviour (as assessed by hypoxia avoidance and plasma cortisol levels) in hatchery bred Atlantic salmon.

I think the results of the study do address the hypothesis at hand, although it seems to be spread across several figures. Thus, I have comments/suggestions regarding data presentation and several major questions regarding the experiments. I also have more minor editorial comments for the authors.

Major questions and comments:

- Seeing that the goal was to investigate a possible relationship between growth and boldness in behaviour, and also that for both are continuous measurements, it should be possible to relate fish weight to frequency of Stayers and Leavers (Fig.4A measurement) and/or to plasma cortisol concentration (Fig.4B measurement) and to look at the correlation between the two variables? This way of data presentation would presumably directly address the hypothesis instead of having weight distribution in one set of figures (Fig.2) and the response variable in 2-3 other figures (Fig.3-4).
- Figure 3C: It's not clear which of data points in here are from fish that were from UM and LM groups. Was there a difference in swimming depth between the two groups? Could the fish classified in the two different groups (if there was a spread) be indicated in different colors to indicate whether there was a difference in swimming depth?
- Figure 4: Due to the distribution of fish weights, is there a size-dependency of plasma cortisol concentration at all?
- Figure 4: Can the authors discuss whether because the frequency of LM fish was lower (Fig.2B) whether that created a bias in selecting fish with certain personalities for the hypoxia avoidance experiment?
- The tentative hypothesis that the authors propose twice in the manuscript that "the occurrence of such contrasts in domesticated lines suggest the tentative hypothesis that successful expression of fast-proactive phenotypes depend on the presence of at least a small fraction of passive copers in the population." This does not seem to be a testable hypothesis. You're always going to have a normal distribution of fish weights and a range of behaviours. Can the authors comment/elaborate on this?
- With the LM being not smoltified and the UM being smoltified, that would indicate large differences in gill morphology that may influence their hypoxia tolerance. Can the authors address whether how that would impact the interpretation of these results?

Minor comments (* indicate my edits):

- Add x-axis labelling into Fig.3A
- Genus and species need to be italicized throughout the manuscript
- "N₂" and "O₂", 2 needs to be in subscript throughout the manuscript
- Figure captions: include sample sizes
- Figure 2A: since the x-axis is not continuous, these points shouldn't be connected with lines
- Figure 4B: Perhaps different color schemes could be used here to match what is used for LM and UM in Fig.4A, since these are all UM fish and no LM fish are present for this dataset
- Page 1, line 35: "This raises the question *of whether* such traits cluster... "
- Page 1, line 45: "There was a higher frequency of In this fast-developing fraction of fish",

higher compared to?

- Page 1, line 56: "... showing consistency between context and over time", is this indicating that behavioural traits displayed by organisms are consistent in different context and over the course of an organisms life-time? Phrasing for POLS unclear.
- Page 1, line 59: "behavioural trait correlation", correlation singular to agree with "is" following.
- Page 2, line 15: "That developmental ratio is related to stress coping styles is further strengthen *by* a study"
- Page 2, line 16: no comma after "O. mykiss, strains"
- Page 2, line 21: "Still, *whether* time to reach..."
- Page 3, line 23: add year to October and November test times
- Page 4, line 46: "... it *became* clear that the fish became increasingly inactive..."
- Page 5, line 14: "This is *consistent* to bimodality of life history traits..."
- Page 5, line 15: "This smoltification process includes a *suite*..."
- Page 5, line 22: "There *was* a higher proportion..."
- Page 5, line 28: Any citation for this previous observation that Stayers would rather die of low O2 than escape?
- Page 5, line 29: "... the individuals have the option of *moving* to other habitats..."
- Page 5, line 37: not sure what you mean here by 'exposed to' higher mortality, "... predicting that proactive/bold personality types *show* higher mortality..."?
- Page 5, line 49: "... aquaculture selects for selecting..", awkward phrasing
- Page 5, lines 51-53: This hypothesis would not be testable and also seems to be a truism. You're always going to have a normal distribution of fish weights and a range of behaviours. Can the authors comment on this?
- Page 5, line 59 to end of page: This sentence is awkwardly long and requires editing: "In line with this,....a sit and wait coping strategy".
- Page 6, line 5-6: This sentence requires a citation: "Generally, the proactive behavioural profile... characterized by a high level of active avoidance together with a low HPI axis reactivity ..."
- Page 6, line 17: "The present *study* shows that active avoidance to hypoxia is associated..."
- Page 6, line 20: "... lends support to a relationship between", on comma after "between"
- Page 6, line 22: "... sorting fish *with* respect to hypoxia avoidance"
-

Review form: Reviewer 2

Is the manuscript scientifically sound in its present form?

Yes

Are the interpretations and conclusions justified by the results?

Yes

Is the language acceptable?

Yes

Is it clear how to access all supporting data?

Not Applicable

Do you have any ethical concerns with this paper?

No

Have you any concerns about statistical analyses in this paper?

No

Recommendation?

Accept with minor revision (please list in comments)

Comments to the Author(s)

Overall I found this paper interesting and easily readable. I thought that the authors outlined the need for further investigation rather well, but it could benefit from a more explicit hypothesis statement in the introduction.

There were a few typos throughout, and I would recommend reading through and fixing as they occur.

I would recommend a justification for using 35g as the split between the LM and UM groups. You noted in the study that there were a limited number of LM individuals in the control treatment that impaired your ability to conduct all statistical tests. Is there a biological or observational reason for this cutoff, or simply numeric?

Decision letter (RSOS-181859.R0)

30-Nov-2018

Dear Professor Damsgård,

The editors assigned to your paper ("Proactive avoidance behaviour and pace-of-life syndrome in Atlantic salmon") have now received comments from reviewers. We would like you to revise your paper in accordance with the referee and Associate Editor suggestions which can be found below (not including confidential reports to the Editor). Please note this decision does not guarantee eventual acceptance.

Please submit a copy of your revised paper before 23-Dec-2018. Please note that the revision deadline will expire at 00.00am on this date. If we do not hear from you within this time then it will be assumed that the paper has been withdrawn. In exceptional circumstances, extensions may be possible if agreed with the Editorial Office in advance. We do not allow multiple rounds of revision so we urge you to make every effort to fully address all of the comments at this stage. If deemed necessary by the Editors, your manuscript will be sent back to one or more of the original reviewers for assessment. If the original reviewers are not available, we may invite new reviewers.

- Data accessibility

If you wish to submit your supporting data or code to Dryad (<http://datadryad.org/>), or modify your current submission to dryad, please use the following link:
<http://datadryad.org/submit?journalID=RSOS&manu=RSOS-181859>

- Competing interests

- Authors' contributions

- Acknowledgements

- Funding statement

Please note that Royal Society Open Science charge article processing charges for all new submissions that are accepted for publication. Charges will also apply to papers transferred to Royal Society Open Science from other Royal Society Publishing journals, as well as papers submitted as part of our collaboration with the Royal Society of Chemistry (<http://rsos.royalsocietypublishing.org/chemistry>). If your manuscript is newly submitted and subsequently accepted for publication, you will be asked to pay the article processing charge, unless you request a waiver and this is approved by Royal Society Publishing. You can find out more about the charges at <http://rsos.royalsocietypublishing.org/page/charges>. Should you have any queries, please contact openscience@royalsociety.org.

on behalf of Dr Michael Tobler (Associate Editor) and Professor Kevin Padian (Subject Editor)
openscience@royalsociety.org

Associate Editor's comments (Dr Michael Tobler):

We have received the feedback from two reviewers that agreed on the merits of the paper. Before the paper can be accepted for publications, the authors should address the constructive feedback from reviewer 1.

Comments to Author:

Reviewers' Comments to Author:

Reviewer: 1

Comments to the Author(s)

I have reviewed the manuscript "Proactive avoidance behaviour and pace of life syndrome in Atlantic salmon". The goal of this study was to investigate whether there is a relationship between growth and development rate and boldness in behaviour (as assessed by hypoxia avoidance and plasma cortisol levels) in hatchery bred Atlantic salmon.

I think the results of the study do address the hypothesis at hand, although it seems to be spread across several figures. Thus, I have comments/suggestions regarding data presentation and several major questions regarding the experiments. I also have more minor editorial comments for the authors.

Major questions and comments:

- Seeing that the goal was to investigate a possible relationship between growth and boldness in behaviour, and also that for both are continuous measurements, it should be possible to relate fish weight to frequency of Stayers and Leavers (Fig.4A measurement) and/or to plasma cortisol concentration (Fig.4B measurement) and to look at the correlation between the two variables? This way of data presentation would presumably directly address the hypothesis instead of having weight distribution in one set of figures (Fig.2) and the response variable in 2-3 other figures (Fig.3-4).

- Figure 3C: It's not clear which of data points in here are from fish that were from UM and LM groups. Was there a difference in swimming depth between the two groups? Could the fish classified in the two different groups (if there was a spread) be indicated in different colors to indicate whether there was a difference in swimming depth?
 - Figure 4: Due to the distribution of fish weights, is there a size-dependency of plasma cortisol concentration at all?
 - Figure 4: Can the authors discuss whether because the frequency of LM fish was lower (Fig.2B) whether that created a bias in selecting fish with certain personalities for the hypoxia avoidance experiment?
 - The tentative hypothesis that the authors propose twice in the manuscript that "the occurrence of such contrasts in domesticated lines suggest the tentative hypothesis that successful expression of fast-proactive phenotypes depend on the presence of at least a small fraction of passive copers in the population." This does not seem to be a testable hypothesis. You're always going to have a normal distribution of fish weights and a range of behaviours. Can the authors comment/elaborate on this?
 - With the LM being not smoltified and the UM being smoltified, that would indicate large differences in gill morphology that may influence their hypoxia tolerance. Can the authors address whether how that would impact the interpretation of these results?
- Minor comments (* indicate my edits):
- Add x-axis labelling into Fig.3A
 - Genus and species need to be italicized throughout the manuscript
 - "N2" and "O2", 2 needs to be in subscript throughout the manuscript
 - Figure captions: include sample sizes
 - Figure 2A: since the x-axis is not continuous, these points shouldn't be connected with lines
 - Figure 4B: Perhaps different color schemes could be used here to match what is used for LM and UM in Fig.4A, since these are all UM fish and no LM fish are present for this dataset
 - Page 1, line 35: "This raises the question *of whether* such traits cluster... "
 - Page 1, line 45: "There was a higher frequency of In this fast-developing fraction of fish", higher compared to?
 - Page 1, line 56: "... showing consistency between context and over time", is this indicating that behavioural traits displayed by organisms are consistent in different context and over the course of an organisms life-time? Phrasing for POLS unclear.
 - Page 1, line 59: "behavioural trait correlation", correlation singular to agree with "is" following.
 - Page 2, line 15: "That developmental ratio is related to stress coping styles is further strengthen *by* a study"
 - Page 2, line 16: no comma after "O. mykiss, strains"
 - Page 2, line 21: "Still, *whether* time to reach..."
 - Page 3, line 23: add year to October and November test times
 - Page 4, line 46: "... it *became* clear that the fish became increasingly inactive..."
 - Page 5, line 14: "This is *consistent* to bimodality of life history traits..."
 - Page 5, line 15: "This smoltification process includes a *suite*..."
 - Page 5, line 22: "There *was* a higher proportion..."
 - Page 5, line 28: Any citation for this previous observation that Stayers would rather die of low O2 than escape?
 - Page 5, line 29: "... the individuals have the option of *moving* to other habitats..."
 - Page 5, line 37: not sure what you mean here by 'exposed to' higher mortality, "... predicting that proactive/bold personality types *show* higher mortality..."?
 - Page 5, line 49: "... aquaculture selects for selecting..", awkward phrasing
 - Page 5, lines 51-53: This hypothesis would not be testable and also seems to be a truism. You're always going to have a normal distribution of fish weights and a range of behaviours. Can the authors comment on this?
 - Page 5, line 59 to end of page: This sentence is awkwardly long and requires editing: "In line with this,.....a sit and wait coping strategy".

- Page 6, line 5-6: This sentence requires a citation: "Generally, the proactive behavioural profile... characterized by a high level of active avoidance together with a low HPI axis reactivity ..."
- Page 6, line 17: "The present *study* shows that active avoidance to hypoxia is associated..."
- Page 6, line 20: "... lends support to a relationship between", on comma after "between"
- Page 6, line 22: "... sorting fish *with* respect to hypoxia avoidance"
-

Reviewer: 2

Comments to the Author(s)

Overall I found this paper interesting and easily readable. I thought that the authors outlined the need for further investigation rather well, but it could benefit from a more explicit hypothesis statement in the introduction.

There were a few typos throughout, and I would recommend reading through and fixing as they occur.

I would recommend a justification for using 35g as the split between the LM and UM groups. You noted in the study that there were a limited number of LM individuals in the control treatment that impaired your ability to conduct all statistical tests. Is there a biological or observational reason for this cutoff, or simply numeric?

Author's Response to Decision Letter for (RSOS-181859.R0)

See Appendix A.

RSOS-181859.R1 (Revision)

Review form: Reviewer 2

Is the manuscript scientifically sound in its present form?

Yes

Are the interpretations and conclusions justified by the results?

Yes

Is the language acceptable?

Yes

Is it clear how to access all supporting data?

Not Applicable

Do you have any ethical concerns with this paper?

No

Have you any concerns about statistical analyses in this paper?

No

Recommendation?

Accept with minor revision (please list in comments)

Comments to the Author(s)

The revisions greatly improve the clarity of hypotheses and how this study fits into the broader scope of the field. A few comments are listed below:

Page 8, Line 15: "where proactive individuals are being characterized...." being should be removed

Page 8, Line 16-20: Reword the sentence that begins "In addition, the stress response in....". In its current state, it is a bit confusing, especially the portion about the low HPA or HPI axis. I know what it means, just rework for clarity.

Page 15, first sentence: "... reactivity axis of the pro- and reactive coping styles /is/ ;ess rigid tha(n) previously though."

Page 15, final sentence of conclusion: occurring should be changed to either "that is occurring" or "that occurs"

Decision letter (RSOS-181859.R1)

15-Feb-2019

Dear Professor Damsgård:

On behalf of the Editors, I am pleased to inform you that your Manuscript RSOS-181859.R1 entitled "Proactive avoidance behaviour and pace-of-life syndrome in Atlantic salmon" has been accepted for publication in Royal Society Open Science subject to minor revision in accordance with the referee suggestions. Please find the referees' comments at the end of this email.

The reviewers and Subject Editor have recommended publication, but also suggest some minor revisions to your manuscript. Therefore, I invite you to respond to the comments and revise your manuscript.

- Ethics statement

- Data accessibility

It is a condition of publication that all supporting data are made available either as supplementary information or preferably in a suitable permanent repository. The data accessibility section should state where the article's supporting data can be accessed. This section should also include details, where possible of where to access other relevant research materials

such as statistical tools, protocols, software etc can be accessed. If the data has been deposited in an external repository this section should list the database, accession number and link to the DOI for all data from the article that has been made publicly available. Data sets that have been deposited in an external repository and have a DOI should also be appropriately cited in the manuscript and included in the reference list.

If you wish to submit your supporting data or code to Dryad (<http://datadryad.org/>), or modify your current submission to dryad, please use the following link:
<http://datadryad.org/submit?journalID=RSOS&manu=RSOS-181859.R1>

- **Competing interests**

- **Authors' contributions**

- **Acknowledgements**

- **Funding statement**

Because the schedule for publication is very tight, it is a condition of publication that you submit the revised version of your manuscript before 24-Feb-2019. Please note that the revision deadline will expire at 00.00am on this date. If you do not think you will be able to meet this date please let me know immediately.

on behalf of Dr Michael Tobler (Associate Editor) and Kevin Padian (Subject Editor)
openscience@royalsociety.org

Associate Editor Comments to Author (Dr Michael Tobler):

The revised version of the manuscript addressed the reviewers concerns, and this paper can be accepted for publication upon minor revisions.

Reviewer comments to Author:

Reviewer: 2

Comments to the Author(s)

The revisions greatly improve the clarity of hypotheses and how this study fits into the broader scope of the field. A few comments are listed below:

Page 8, Line 15: "where proactive individuals are being characterized...." being should be removed

Page 8, Line 16-20: Reword the sentence that begins "In addition, the stress response in.....". In its current state, it is a bit confusing, especially the portion about the low HPA or HPI axis. I know what it means, just rework for clarity.

Page 15, first sentence: "... reactivity axis of the pro- and reactive coping styles /is/ ;ess rigid tha(n) previously though."

Page 15, final sentence of conclusion: occurring should be changed to either "that is occurring" or "that occurs"

Author's Response to Decision Letter for (RSOS-181859.R1)

See Appendix B.

Decision letter (RSOS-181859.R2)

18-Feb-2019

Dear Professor Damsgård,

I am pleased to inform you that your manuscript entitled "Proactive avoidance behaviour and pace-of-life syndrome in Atlantic salmon" is now accepted for publication in Royal Society Open Science.

on behalf of Dr Michael Tobler (Associate Editor) and Professor Kevin Padian (Subject Editor)
openscience@royalsociety.org

Appendix A

Feedback to the reviews of ID RSOS-181859, 14 Dec. 2018

Reviewer 1

1. *I have reviewed the manuscript "Proactive avoidance behaviour and pace of life syndrome in Atlantic salmon". The goal of this study was to investigate whether there is a relationship between growth and development rate and boldness in behaviour (as assessed by hypoxia avoidance and plasma cortisol levels) in hatchery bred Atlantic salmon.*

I think the results of the study do address the hypothesis at hand, although it seems to be spread across several figures. Thus, I have comments/suggestions regarding data presentation and several major questions regarding the experiments. I also have more minor editorial comments for the authors.

Our response: We have looked into the suggestions made by the reviewer.

Major questions and comments:

2. *Seeing that the goal was to investigate a possible relationship between growth and boldness in behaviour, and also that for both are continuous measurements, it should be possible to relate fish weight to frequency of Stayers and Leavers (Fig.4A measurement) and/or to plasma cortisol concentration (Fig.4B measurement) and to look at the correlation between the two variables? This way of data presentation would presumably directly address the hypothesis instead of having weight distribution in one set of figures (Fig.2) and the response variable in 2-3 other figures (Fig.3-4).*

Our response: The issue raised by the reviewer is interesting, but it is not a question of continuous measurements and thus not an expected correlation between size and any variable. Any behavioural differences that can be explained in terms of unimodal size differences may simply be part of a proximate physiological explanations, rather than POLS. In anadromous salmonid fishes the life history choice of sea migration related to the development the year before smoltification. In line with this, the seawater challenge revealed the fast-growing upper mode fish smoltified, while the lower mode group did not. It is well known that growth thus is a consequence of the life history decision to become a smolt, and smoltification *per se* is not a consequence of size. Accordingly, we treated UM and LM as grouping variables. Like vice, the hypoxia test revealed two contrasting behavioural responses to this challenge; leavers and stayers. A confinement stress test confirmed that

these two contrasting behavioural reactions to hypoxia corresponded to the proactive and reactive stress coping styles respectively. Thus, the bimodality of growth/developmental trajectories and the contrasting stress coping styles suggests that the data should be treated as categorical rather than continuous.

We do however see from the reviewers' concerns that the aim of the study needs to be clarified. To further clarify this, we have changed the last paragraph in the introduction to clarify that it is a question of bimodality due to the growth pattern based on the life history decision to reach smoltification.

3. *Figure 3C: It's not clear which of data points in here are from fish that were from UM and LM groups. Was there a difference in swimming depth between the two groups? Could the fish classified in the two different groups (if there was a spread) be indicated in different colors to indicate whether there was a difference in swimming depth?*

Our response: We see the challenge here to understand the complexity in hypoxia responses, and have improved both the Materials and Methods (2.2), Results (3.2) and Figure legends to clarify this. Our study provides a very detailed and consistent way of measuring the responses to a decline in oxygen, and we felt it was necessary to show the changes in oxygen (Fig. 3 a) and swimming depth (Fig. 3 b) from 80 minutes before to 80 minutes after the onset of hypoxia. These two figures do however not show the relationship between the saturation and the swimming depth, and the data point were thus presented in Fig. 3 C, because it is not the time *per se* in the system that trigger the behavioural changes, but a threshold oxygen saturation. The data points in Fig. 3 C are not fish individuals, but measurements of swimming depths (based on triggering of infrared sensors), and it is thus not possible to separate between the two groups. It is however a very good idea to develop such method in the future. The identification of the individual fish could first be done when they left hypoxia and the tag number could be recorded by the PitTag antenna.

4. *Figure 4: Due to the distribution of fish weights, is there a size-dependency of plasma cortisol concentration at all?*

Our response: The main difference in the plasma cortisol was between the two groups of UM and LM, and there was only a weak ($R^2=0.03$) relationship in the model. It is thus not a size-dependency in our data, but a group dependency.

5. *Figure 4: Can the authors discuss whether because the frequency of LM fish was lower (Fig.2B) whether that created a bias in selecting fish with certain personalities for the hypoxia avoidance experiment?*

Our response: All fish that were used in all tests were un-selected and due to this and a very low mortality since hatching we might expect the fish to represent a whole un-biased population. The hypoxia tests were done a time period before the stress tests in order to avoid effects of the tests, and the stress tests was done in un-selected groups independent of the result in the hypoxia test. This protocol was used to avoid biased results, but the downside of that is of course that the numbers of individual in the end result will differ. The chi-square test does not require equal group sizes, so a lower number of individuals in the LM group are not expected to induce a type I error.

6. *The tentative hypothesis that the authors propose twice in the manuscript that “the occurrence of such contrasts in domesticated lines suggest the tentative hypothesis that successful expression of fast-proactive phenotypes depend on the presence of at least a small fraction of passive copers in the population.” This does not seem to be a testable hypothesis. You’re always going to have a normal distribution of fish weights and a range of behaviours. Can the authors comment/elaborate on this?*

Our response: We understand the concern of the reviewer, and this statement was not meant as conclusions from our study, but rather as a hypothesis that might be tested in future studies. Our study demonstrates the occurrence of behavioural traits linked to distinct developmental/growth trajectories, but it is still an open question why such differences exist, and how domestication in aquaculture should take that into account. We have omitted the statement from the Conclusion and changed it in the Discussion to avoid misunderstandings.

7. *With the LM being not smoltified and the UM being smoltified, that would indicate large differences in gill morphology that may influence their hypoxia tolerance. Can the authors address whether how that would impact the interpretation of these results?*

Our response: That is a good question that unfortunately cannot be tested in our setup, but needs a completely different physiological approach. However, size differences could not explain the hypoxia response, but there might of course be other, not size dependent, physiological mechanisms explaining why the UM group have a lower threshold for hypoxia. In the gills, the smoltification process mostly affects the development of the chloride cells, and not the gill morphology. A study of Jenjan *et al.* 2013 (Respiratory function in common carp..., *Animal Behaviour* 85, 1245-1249) demonstrated a larger gill surface in proactive fish. Since the proactive fish were the first to leave the hypoxia, a different oxygen threshold cannot be explained as a larger gill surface. Such mechanisms have not been studied in salmon smolts, and we cannot speculate based upon our study.

Minor comments (* indicate my edits):

8. *Add x-axis labelling into Fig.3A*

Our response: The x-axis in Fig. 3 a was omitted because it is the same as in 3 b, and it is stated in the legend that it is the same. Based on the comments we have included it again.

9. *Genus and species need to be italicized throughout the manuscript*

10. *"N2" and "O2", 2 needs to be in subscript throughout the manuscript*

Our response: Both point 9 and 10 seems to be happen when the document is uploaded to RSOS, including some letters in the title. We will correct it again, and hopefully it will be correct in the next upload.

11. *Figure captions: include sample sizes*

Our response: We have included sample size in the text, e.g. in connection with comparisons and statistical tests, and generally think that a full list of sample sizes in the figure captions are making the legends less readable. To meet the reviewer's comment, we have however included the sample sizes when possible in the figure captions.

12. *Figure 2A: since the x-axis is not continuous, these points shouldn't be connected with lines*

Our response: We do not disagree with the reviewer, and included the lines in order to make the figure easier to understand. We have omitted the lines and changed to a bar figure.

13. *Figure 4B: Perhaps different color schemes could be used here to match what is used for LM and UM in Fig.4A, since these are all UM fish and no LM fish are present for this dataset*

Our response: It is right that Fig. 4 b is only UM fish (as stated in the figure legend), but the use of black and grey is consistent in both figures, and we have used black/white figures throughout the whole manuscript.

14. *Page 1, line 35: "This raises the question *of whether* such traits cluster... "*

Our response: Corrected.

15. Page 1, line 45: *“There was a higher frequency of In this fast-developing fraction of fish”, higher compared to?*

Our response: Corrected.

16. Page 1, line 56: *“... showing consistency between context and over time”, is this indicating that behavioural traits displayed by organisms are consistent in different context and over the course of an organisms life-time?*

Our response: The question raised by the reviewer is interesting and relevant, but beyond the scope of this study, and in this part of the text we are using the normal definitions of this traits. However, the majority of studies regarding animal personality have been performed in adult animals, and recently questions have been raised regarding the existence of such trait correlations early in development and the relation to personality traits expressed later in life (For references, see Groothuis & Trillmich 2011. Unfolding personalities: The importance of studying ontogeny. *Dev Psychobiol* 53(6): 641–655). We used the common definition stated in Reale et al. 2010. Personality and the emergence of the pace-of-life syndrome concept at the population level. *Philosophical Transactions of the Royal Society B: Biological Sciences*. 365: 4051–4063. They do not specify if they consider the constancy over time should be life long, but discuss the genetic contribution to the behavioural profile of an individual, which can imply that individual variation in behaviour should show constancy over a life time. To what extent a behavioural trait is life-long will be a bit out of scope of this paper.

17. *Phrasing for POLS unclear.*

Our response: We are not quite sure what the reviewer means, but we have included an explanation of the term in the introduction

18. Page 1, line 59: *“behavioural trait correlation”, correlation singular to agree with “is” following.*

Our response: Corrected.

19. Page 2, line 15: *“That developmental ratio is related to stress coping styles is further strengthen *by* a study”*

Our response: Corrected.

20. Page 2, line 16: *no comma after “O. mykiss, strains”*

Our response: Corrected.

21. Page 2, line 21: *"Still, ***whether*** time to reach..."*

Our response: Corrected.

22. Page 3, line 23: *add year to October and November test times*

Our response: Corrected.

23. Page 4, line 46: *"... it ***became*** clear that the fish became increasingly inactive..."*

Our response: Corrected.

24. Page 5, line 14: *"This is ***consistent*** to bimodality of life history traits..."*

Our response: Corrected.

25. Page 5, line 15: *"This smoltification process includes a ***suite***..."*

Our response: Corrected.

26. Page 5, line 22: *"There ***was*** a higher proportion..."*

Our response: Corrected.

27. Page 5, line 28: *Any citation for this previous observation that Stayers would rather die of low O2 than escape?*

Our response: We have not been able to find any published data on this, but it is our experience that hypoxia tests should not go below 25% in salmonids, as this may be harmful for the fish, and will not change the result as the passive fish rather wait and thus run a risk of dying of low oxygen. We have clarified this in the manuscript.

28. Page 5, line 29: *"... the individuals have the option of ***moving*** to other habitats..."*

Our response: Corrected.

29. Page 5, line 37: *not sure what you mean here by 'exposed to' higher mortality, "... predicting that proactive/bold personality types ***show*** higher mortality..."?*

Our response: Corrected.

30. Page 5, line 49: "... aquaculture selects for selecting..", awkward phrasing

Our response: Corrected.

31. Page 5, lines 51-53: This hypothesis would not be testable and also seems to be a truism. You're always going to have a normal distribution of fish weights and a range of behaviours. Can the authors comment on this?

Our response: The problem is changed according to point 6.

32. Page 5, line 59 to end of page: This sentence is awkwardly long and requires editing: "In line with this,...a sit and wait coping strategy".

Our response: A full stop was missing, and has been corrected.

33. Page 6, line 5-6: This sentence requires a citation: "Generally, the proactive behavioural profile... characterized by a high level of active avoidance together with a low HPI axis reactivity ..."

Our response: We have included a review citation.

34. Page 6, line 17: "The present *study* shows that active avoidance to hypoxia is associated..."

Our response: Corrected.

35. Page 6, line 20: "... lends support to a relationship between", on comma after "between"

Our response: Corrected.

36. Page 6, line 22: "... sorting fish *with* respect to hypoxia avoidance"

Our response: Corrected.

Reviewer 2

37. Overall I found this paper interesting and easily readable. I thought that the authors outlined the need for further investigation rather well, but it could benefit from a more explicit hypothesis statement in the introduction.

Our response: We have changed the last part of the introduction so the hypotheses are clarified.

38. *There were a few typos throughout, and I would recommend reading through and fixing as they occur.*

Our response: We have looked carefully at the manuscript in order to take out typos, and some of them have been kindly identified by reviewer 1.

39. *I would recommend a justification for using 35g as the split between the LM and UM groups. You noted in the study that there were a limited number of LM individuals in the control treatment that impaired your ability to conduct all statistical tests. Is there a biological or observational reason for this cutoff, or simply numeric?*

Our response: The separation between LM and UM are numerical, and there are no non-invasive biological ways to separate clearly between the two developmental groups. There might of course be an overlap between the two groups, but in this case the two modals were easily separated at 35 grams. This cut-off is indicated with a dotted line in Fig. 1 b.

Appendix B

Feedback to the reviews of ID RSOS-181859, 18 Feb. 2019

Reviewer: 2

The revisions greatly improve the clarity of hypotheses and how this study fits into the broader scope of the field. A few comments are listed below:

1. Page 8, Line 15: "where proactive individuals are being characterized...." being should be removed

Our response: The part of the sentence has been removed.

2. Page 8, Line 16-20: Reword the sentence that begins "In addition, the stress response in.....". In its current state, it is a bit confusing, especially the portion about the low HPA or HPI axis. I know what it means, just rework for clarity.

Our response: We agree that the sentence was difficult to read and have separated it into two sentences.

3. Page 15, first sentence: "... reactivity axis of the pro- and reactive coping styles /is/ ;ess rigid tha(n) previously though."

Our response: The verb had been deleted from the sentence with a mistake, and has now been included again.

4. Page 15, final sentence of conclusion: occurring should be changed to either "that is occurring" or "that occurs"

Our response: The sentence have been changed to "that occurs"